# An Analysis of Rural Households’ Earthquake-Resistant Construction Behavior: Evidence from Pingliang and Yuxi, China

**DOI:** 10.3390/ijerph17239079

**Published:** 2020-12-04

**Authors:** Manqing Wu, Guochun Wu

**Affiliations:** Institute of Geophysics, China Earthquake Administration, Beijing 100081, China; wumanqing18@mails.ucas.ac.cn

**Keywords:** rural housing safety, policy subsidy, self-built houses, earthquake-resistant structures, earthquake preparedness

## Abstract

Due to the lack of earthquake-resistant rural houses, small and medium earthquakes caused massive casualties in rural China. In 2004, China began implementing the Earthquake Rural Housing Safety Project Policy (ERHSPP) to reduce earthquake losses, mainly promoting the adoption of earthquake-resistant structures in rural residents’ self-built houses through subsidies, training of construction craftsmen, and provision of earthquake-resistant housing drawings. We conducted a field survey, collecting 1169 rural households in Pingliang, Gansu Province, and 1501 rural households in Yuxi, Yunnan Province, China. We studied Earthquake-Resistant Construction Behaviors (E-RCB) by the logistic and the ordered logistic regression models. Results show that government housing subsidy promotes E-RCB of rural households; E-RCB was affected by ERHSPP, positively correlated with economic status and housing earthquake damage; E-RCB was negatively correlated with structure danger, house age, and earthquake experience; and housing earthquake damage, not earthquake experience, strikingly promoted E-RCB in rural China. The results could provide suggestions in communication risk strategies for the government. We suggest the local government should promote local acceptable disaster propaganda, provide hierarchical housing subsidies, pay attention to housing seismic supervision, publish earthquake-resistant housing design drawings, and conduct more earthquake-resistant technical training for rural craftsmen.

## 1. Introduction

In China, 650 million rural people live in earthquake-hazardous areas where the Earthquake Intensity Degree ≥ 6. However, the lack of supervision and guidance during rural houses’ construction caused huge losses after the earthquake. From 2001 to 2005, 60 disastrous earthquakes occurred in mainland China, affecting approximately 7684 million people, 144,300 km^2^ area, and 9496 billion yuan loss [1]. In 2003, three earthquakes of magnitude ≥5 happened in Jiashi County, Xinjiang Province, causing 1502 billion yuan loss.

To enhance the seismic performance of rural self-built houses, ERHSPP was implemented in Xinjiang in 2004 as a pilot, and it was officially promoted throughout China in 2007 in the Opinions on the Implementation of Earthquake Safety Project for Rural Residential Buildings [2,3]. According to China Earthquake Administration (CEA) disaster reduction plans and regulations from 2007 to 2019, out of 31 provinces in Mainland China, 100% strengthened rural earthquake reduction education; 96.8% built model rural houses; 96.8% developed practical seismic technical services; 87.1% formulated local rural house construction plans; 80.6% of provinces supervised rural house site selection, avoidance, and construction quality; 74.2% of provinces trained skilled craftsmen to master the seismic knowledge; 71.0% of provinces publicized measures and experiences in rural housing policies; 71.0% of provinces provided policy subsidies; and 64.5% of provinces established long-term service organizations.

ERHSPP has been successful in improving the earthquake resistance of rural houses. None of the newly built rural earthquake-resistant houses in Xinjiang province after 2004 were damaged after more than 60 earthquakes of magnitude ≥5 [3]. Therefore, studying how ERHSPP promotes E-RCB is of great significance in improving rural earthquake safety. In practice, ERHSPP not only refers to the policies of the earthquake department, but also includes Chinese government policies related to rural poverty alleviation. According to the 2014 Central Agricultural Document No. 1, Chinese government implemented ERHSPP in earthquake-hazardous areas, and county-level governments integrated and coordinated the use of poverty alleviation subsidies to promote ERHSPP [4]. The original purpose of ERHSPP was to reduce disaster losses in rural areas. Later, it was used as a poverty alleviation policy in rural areas. Although the name of the policy has changed, the Chinese government has always been concerned about rural housing safety and reducing rural disaster vulnerability to achieve sustainable development goals. This article refers to the policies on rural housing safety in mainland China after 2004 as the Earthquake Rural Housing Safety Project Policy (ERHSPP).

Prior research on household disaster preparedness discussed psychological aspects and adaptive behaviors. We would like to explore the factors affecting actual behaviors. The Government housing subsidy in this article refers specifically to the special housing subsidies in ERHSPP, The Earthquake-Resistant Construction Behaviors (E-RCB) refer specifically to used seismic drawing, made seismic patands, and chosen qualified contractors when building the house. This paper adopts logistic and ordered logistic regression to quantitatively analyze E-RCB, ERHSPP effect, and other aspects to investigate:

Did government housing subsidy promote rural household E-RCB or not?

Are government project’s effect, house risk, economic level, and earthquake experience related to E-RCB or not?

E-RCB in Yuxi and Pingliang, China, are conducted as the case study through questionnaire surveys. This article includes five parts: Section 1 is the introduction, related literature review, and research hypotheses. Section 2 introduces the survey study, the logistic regression models, and the ordered logistic regression models. Section 3 explains the results. Section 4 discusses these results. The fifth section lists some suggestions.

### 1.1. Theoretical Basis and Variable Measure

Existing research on household disaster preparedness discussed psychological aspects and adjustment behavior. Psychological research on household disaster preparedness discussed the factors that influence individual risk perceptions. Differences in individual perceptions of disaster risk are related to gender, age, education, socio-economic status, household income, housing economic value, the risk level of the residence, the number of children, individual responsibility for taking preventive measures, and psychosocial factors [5,6,7,8]. On the other hand, existing research described the individual’s feelings and thought processes in processing disaster information, examining the individual’s psychological processes rather than the actual actions [9,10].

In terms of adjustment behaviors, studies discussed household and individual disaster preparedness and adjustment behaviors, mainly including the apportionment of economic losses such as earthquake insurance [11,12,13,14,15], emergency kits and manuals reserves [16,17], seismic reinforcement or fixation of existing furniture [18,19,20], and periodic emergency escape drills [21,22].

Another possible research direction is to analyze the relationship between government project effect and household disaster preparedness. According to the practice of ERHSPP, this article defines “Earthquake-Resistant Construction Behaviors (E-RCB)” as the use of seismic drawing, seismic patands, and qualified contractors when rural households build their own houses. The hypotheses of E-RCB are as follows.

### 1.2. Government Project Effect

Scholars have studied the impact on the government engineering projects, and whether the impact on residents’ behavior is positive or negative. There are different opinions.

The government’s dissemination of disaster preparedness knowledge to encourage people to understand local disaster risk information is considered to be related to residents’ disaster preparedness behavior. The government’s disaster mitigation plan can promote residents to understand their location’s disaster risk [23,24]. In earthquake-hazardous areas, earthquake departments often publicize local seismic fortification intensity and formulate local emergency countermeasures to affect residents’ risk perception. Lindell confirmed that individual risk perception plays a central role in promoting family disaster preparedness behavior, and significantly positively correlates with individual intentions and actual earthquake preparedness measures [25,26]. Paton found that the government’s disaster education measures promoted public risk perception and improved individual disaster preparedness behavior [27]. Parajuli adopted a multidisciplinary participatory approach and conducted a local level resilience assessment. It was considered that Citizen Disaster Science Education (CDSE) “will help residences understand disaster risks properly, which will guide residences to make decisions to reduce exposure” [28,29].

Residents’ government trust is considered to be related to residents’ disaster preparedness. Han took Yushu earthquake survivors as samples, and pointed out that the two self-evaluation indicators of individual government trust and household disaster preparedness were significantly negatively correlated [30]. Additionally, Cheng surveyed 417 residents that live in earthquake-hazardous areas, and confirmed the trust of government information and policy influence the acceptation intention for seismic mitigation policies [31].

Special government housing subsidies and similar economic subsidies are considered related to residents’ disaster preparedness behavior. Taylan found linking house maintenance credits with adoption of risk mitigation measures, motivating homeowners for taking earthquake preparedness measures in Turkey [32]. On the other hand, Wu’s analysis of Yuxi ERHSPP data shows that government subsidy standards were not linked to household income. In practice, the more affordable households are, the more likely they are to receive government subsidies [33].

Additionally, households may also passively rely on government actions. Wu’s research on rural communities in Kaohsiung City found that after the implementation of the disaster reduction plan, 50% of the communities passively rely on the assistance of the government and experts, and only about 1/6 of the communities have mobilized more materials for disaster reduction preparations with government policy support [34]. Tian used rural residents in Chuxiong Prefecture as a sample and found that individual catastrophe insurance needs are negatively correlated with the government’s engineering prevention behavior [35].

Therefore, previous studies have different views on the impact of government engineering measures on residents’ behavior. This article provides the following hypotheses:

**Hypothesis** **1a** **(H1a).** 
*Government project effect and E-RCB are positively correlated.*


**Hypothesis** **1b** **(H1b).** 
*Government project effect and E-RCB are negatively correlated.*


**Hypothesis** **1c** **(H1c).** 
*Government housing subsidy and E-RCB are positively correlated.*


**Hypothesis** **1d** **(H1d).** 
*Government housing subsidy and E-RCB are negatively correlated.*


### 1.3. House Risk

The comparison between the new and old built rural houses can reflect the effect of ERHSPP. We considered house age to be related to the seismic performance of the house. According to the Second National Agricultural Census in 2006 [36], in rural China, only 6.0% of the building structures of households’ self-built houses were reinforced concrete, brick–concrete and brick–wood accounted for 83.7%, and bamboo grass adobe accounted for 9.6%. The three types of house structures have different risks. According to the statistics of Ming, in northwestern Yunnan’s 25 destructive earthquakes from 1992 to 2015, the damage ratio of frame, brick–concrete, brick-–wood, and civil-timber buildings increased in order [37]. Zhong calculated the average earthquake damage index and confirmed that the seismic performance of various rural house structures in the northwest region increases in the order of recent brick–concrete, old brick–concrete, brick–timber, and adobe wall civil structures [38].

By the Standard for Seismic Appraisal of Buildings (GB 50023–2009) [39], the expected subsequent service life of houses and applicable seismic appraisal methods are divided according to the house’s completed year. The later the house completed, the longer the subsequent service life, the higher the applicable seismic appraisal standard. For houses that use seismic design codes, existing buildings built in the 1980s and before are more damaged than those built after the 1990s when they encounter the same earthquake impact. This article assumes that houses with higher structural danger and older houses need more E-RCB. That is, houses with more E-RCB have lower structural danger and younger houses.

**Hypothesis** **2** **(H2).** 
*Structural danger and house age are negatively related to E-RCB.*


### 1.4. Economic Level

We considered the economic level represented by income directly related to the household earthquake preparedness behavior [40]. Economic difficulties are likely to prevent individuals from turning disaster preparedness awareness into practice. Individuals may understand disaster risks but are unable to prevent them due to economic difficulties. The representative factors causing this paradox are the insufficient amount of adjustable resources and the household’s low financial affordability [41]. Even if an individual shows a preference for disaster preparedness behavior, this motivation may not necessarily be implemented into actual behavior due to practical obstacles such as money, which is unpredictable when the motivation is formed [42,43]. Therefore, this article assumes that the economic level and E-RCB are positively correlated.

**Hypothesis** **3** **(H3).** 
*Economic level and E-RCB are positively correlated.*


### 1.5. Earthquake Experience

Considering that a catastrophe is a small probability event, it may be challenging to motivate households to build strong houses for earthquake preparedness, which is time-consuming and labor-intensive. Previous studies generally believed that the individual’s past disaster experiences such as directly experiencing large-scale destructive earthquakes and experiencing earthquake losses will help them decide how to respond to earthquakes. Compared with individuals who have not personally experienced earthquakes, individuals who have experienced earthquakes have taken more disaster preparedness measures [44,45,46]. Becker also found that other small disaster experiences, life accidents, and word-of-mouth disaster descriptions can alternatively help individuals understand the consequences of disasters and promote disaster preparedness interactions in the community [47]. Therefore, this paper takes the experience of earthquake damage and the loss of housing damage as indicators of earthquake damage experience, assuming earthquake experience and E-RCB are positively correlated.

**Hypothesis** **4** **(H4).** 
*Earthquake experience and E-RCB are positively correlated.*


## 2. Materials and Methods

### 2.1. Introduction to the Research Area

Wu has conducted field surveys on the affordability costs of the seismic fortification of rural houses in Pingliang, Gansu Province, in 2010, and Yuxi, Yunnan Province, in 2011 (Figure 1).

The sample questionnaires collected valid sample information of 2670 rural households, among 1169 samples from Pingliang, Gansu Province, and 1501 samples from Yuxi, Yunnan Province. The sample questionnaires are the data source of this study. We used multi-stage stratified sampling, selecting villages with Earthquake Intensity Degree ≥ 7. Each family chose one person to form a list and stratified according to local population information sampling. Interviewers conducted face-to-face interviews due to the education level of the interviewees. This study uses Stata 15 for data analysis. Two survey sites are shown in Table 1, seismic fortification intensity ≥ 7, which belongs to earthquake-hazardous areas in China, within the range of China’s North–South Seismic Belt. Yuxi City and Pingliang City are relatively poor during the survey, and the local government implemented ERHSPP early. The statistical information of the interviewee is shown in Table 2.

Both places have experienced major earthquakes in 1970 and 2008. The 7.7 Ms Tonghai earthquake on 15 January 1970, caused 15,621 deaths, and 338,456 houses collapsed. The entire area of Yuxi City caused heavy losses due to the 1970 Tonghai earthquake. The 8.0 Ms Wenchuan Earthquake on 12 May 2008 caused 69,227 deaths and severely damaged more than a 100,000 km^2^ area. Yuxi and Pingliang City both belong to the earthquake-affected area.

### 2.2. Dependent Variables

This research mainly analyzes whether government project effect, house risk, economic level, and earthquake experience are related to E-RCB. The descriptive statistics of the variables are shown in Table 3.

The first dependent variable is Policy Subsidy, which reflects received the government housing subsidy when building this house with seismic structure. The questionnaire asked, “Have you received government subsidies for building/purchasing this house?”, 1 for “Yes” and 0 for “No”. 6.03% sample households received special housing subsidies. We only raised this question in Yuxi, and the total sample size for this variable is 1501.

The second dependent variable is Earthquake-Resistant Construction Behaviors (E-RCB), which consists of three measures: used seismic drawing, made seismic patands, and chosen qualified contractors when building the house. 16.93% sample households completed all three measures, 9.78% sample households completed two measures, and 5.54% sample households completed one measure.

### 2.3. Independent Variables

Seismic Drawing: Used seismic drawing when building this house. The questionnaire asked “Is the construction drawings used when building this house?”, “Does the construction drawings consider seismic resistance?”, 1 for “Used” and “Considered”, 0 for “Used” but “Not considered”, and 0 for “Not used”. 15.88% sample households used seismic drawing.

Seismic Patands: Made seismic patands when building this house. The questionnaire asked “Has the foundation of this house been treated?”, 1 for “Treated, there are ground ring beams” and 0 for “Simple treatment” and “No treatment”. 27.75% sample households made seismic patands.

Qualified Contractors: Chosen qualified contractors with seismic construction skills when building this house. The questionnaire asked, “Who built this house?”, 1 for “Qualified construction team” and 0 for “Unqualified construction team” and “Friends and relatives I find myself”. 9.48% sample households chose qualified contractors.

Policy Information: Knew the seismic fortification intensity of the area (or the seismic fortification standard to be achieved when building houses). The questionnaire from Pingliang asked “Do you know the seismic fortification intensity in this area?” and the questionnaire from Yuxi asked “Do you know the seismic fortification standards to be met when building houses in this area?”, 1 for “Known” and 0 for “Never heard of it”. 10.15% sample households knew the seismic fortification intensity.

Government Trust: Recognition degree of “the local government’s earthquake countermeasures are adequate”. The questionnaire asked “How would you rate ‘the local government’s countermeasures to the earthquake are adequate’”; the options “Inconsistent”, “Not very consistent”, “Relatively consistent”, and “Consistent” are assigned a value of 0–3 in turn. 37.42% sample households are relatively consistent with this opinion.

Structural Danger: The corresponding risk of the building structure can be divided into “Ring beam structure or steel frame”, “Brick–concrete or brick–wood”, and “Adobe” concerning Seismic Field Work Part 3: Survey Specifications (GB/T 18208.3-2011) [48]. The hazard levels increase in order, and 0–2 points are assigned in order. The questionnaire asked “The type of house in this house”: 0 for “Ring beam structure column” and “Reinforced concrete frame”, 1 for the note “Brick–concrete” or “Brick–wood” in “Others”, and 2 for “Adobe”. 38.61% of sample houses were brick–concrete or brick–wood structure.

House Age: The questionnaire asked, “When was this house built?” House Age is equal to the survey year minus the house construction year, and the result is in logarithm. The mean of the sample logarithm house age was 2.400.

Annual Income: The questionnaire asked, “How much income did your family have in the past year?” The annual family income is RMB. 5000 yuan, 5000–8000 yuan, 8000–10,000 yuan, 10,000–20,000 yuan, 20,000–30,000 yuan, 30,000–50,000 yuan, 50,000–80,000 yuan, 80,000–100,000 yuan, 100,000–200,000 yuan, 200,000–500,000 yuan, and more than 500,000 yuan are assigned 1–11 points in turn. The mean of sample households’ annual income was within the range of 8000–20,000 yuan.

Earthquake Experience: Experienced 1970 Tonghai or 2008 Wenchuan earthquake. Questionnaire of Pingliang asked “Have you encountered an earthquake disaster so far?”: 1 for “Have experienced an earthquake disaster” and 0 for “Have experienced other disaster “ and “Have not suffered a disaster”. Questionnaire of Yuxi asked “Have you heard of the 1970 Tonghai earthquake?”: 1 for “Experienced personally” and 0 for “Heard of” and “Never heard of”. Out of the sample households, 55.06% experienced the 2008 Wenchuan or 1970 Tonghai disastrous earthquake.

Housing Damage: The house has previously collapsed due to an earthquake. The questionnaire asked “What degree of earthquake disaster did you suffer?”: 1 for “My house completely collapsed” and “My house partially collapsed” and 0 for “No house damage”. Out of the sample households, 30.19% experienced self-housing collapse due to an earthquake.

To conduct hypothesis testing, this article regards the explanatory variable “Earthquake-Resistant Construction Behaviors (E-RCB) “ as an ordinal variable, and establishes an ordered logistic regression model with the expression:(1)$$P\left(y=j\;|x_{i}\right)=\frac{1}{1+e-\left(\alpha +\beta X_{i}\right)}$$

$X_{i}$ (i = 1,2…7) represents the independent variable, and y represents the probability of E-RCB.
(2)$$\mathrm{Logit}\left(P_{j}\right)=\ln\left[\frac{P\left(y\leq j\right)}{y\geq j+1}\right]=\alpha_{j}+\overset{n}{\underset{i=1}{\sum }}\beta_{i}X_{i}$$

$P_{j}$ = (P = j), which represents the probability ratio of item j (j = 0,1,2,3) occurring in E-RCB, and $X_{i}$ (i = 1,2…7) represents the impact of E-RCB, variable $\beta_{i}$ represents the regression coefficient corresponding to $X_{i}$, and $\alpha_{j}$ represents the intercept of the model.

## 3. Results

We aimed to explore whether the policy subsidy has an incentive effect on E-RCB and whether the household received the policy subsidy as a dependent variable. The use of seismic drawings that reflect technical services, the qualified construction team that reflects the training of construction craftsmen, and the seismic construction of self-built houses are used as independent variables. We only raised this question in Yuxi, and the total sample size for this variable is 1501. In Table 4, the logistic regression results of Model A–D show that policy subsidy is positively correlated with the three independent input E-CRB variables, which shows that policy subsidy has a significant promotion effect on E-RCB. Moreover, it is not necessary to complete all three E-RCB to get the policy subsidy. On the whole, policy subsidy has the most significant effect in promoting the seismic patand construction of rural houses.

This paper uses E-RCB as the dependent variable. We tested the interaction effects of the independent variables, and the results showed that there was no significant interaction effect between the independent variables, so we chose to use the following model. Model 1 is the basic model to examine the impact of government project effect on E-RCB. Model 2 adds house variables based on Model 1 to investigate the influence of house risk on E-RCB. Model 3 adds economic variables to Model 2 to examine the impact of the economic level on E-RCB. Model 4 adds experience variables based on model 3 to investigate the impact of earthquake experience on E-RCB. The results of Model 1–4 are shown in Table 5.

The results of Model 1 show that after controlling government trust, compared with households that did not know the local seismic fortification standards, households that know the standards enacted more E-RCB; the increase is about 94.3% ($=e^{0.664}-1$). After the control policy information, the rate of occurrence of more E-RCB increases by about 13.4% ($=e^{0.126}-1$) for every level of government trust raised. Therefore, policy information and government trust have a significant positive effect on E-RCB. H1a–d is supported, and government project effects E-RCB and is positively correlated.

Model 2 adds house variables based on Model 1. The results show that after controlling for other factors, every time the house’s structural danger increases by one level, the occurrence of more E-RCB is reduced by approximately 75.5% ($=1-e^{-1.405}$). After controlling for other factors, compared with houses with relatively early age, the incidence of more E-RCB by households with older houses decreased by about 42.0% ($=1-e^{-0.545}$). Therefore, the risk of a building structure and the house age have a significant negative effect on the E-RCB. H2 is supported, and the risk of the building structure, the house age, and E-RCB are negatively correlated.

Model 3 adds economic level variables based on Model 2. The results show that after controlling for other factors, compared with households with lower annual incomes, the incidence of more E-RCB s in households with higher annual incomes increased by approximately 34.4% ($=e^{0.296}-1$).

Therefore, the economic level has a significant positive effect on E-RCB. H3 is supported, and the economic level and E-RCB are positively correlated.

Model 4 adds experience variables based on Model 3. The results show that after controlling for other factors, compared with households that have not personally experienced the earthquake, the occurrence of more E-RCB of families who have experienced the earthquake is reduced by about 71.3% ($=1-e^{-1.247}$). After controlling for other factors, compared with the households without housing damages, the incidence of more E-RCB to households with housing damages increased by approximately 149.2% ($=e^{0.913}-1$). Therefore, the experience of E-RCB has a significant positive effect, and the loss of housing damage to E-RCB has a significant negative effect. H4 is not supported, and the experience of E-RCB is negatively related to the loss of housing damage and E-RCB. It is positively correlated.

The results of the research are shown in Figure 2. As summarized in Table 6, the results of this study support H1a, H1c, H2, and H3; partially support H4; and contradict H1b and H1d.

## 4. Discussion

This paper uses the ordered logistic regression model to explore the influence of policy factors on E-RCB, and the influence of other non-policy factors on E-RCB, found:

E-RCB is significantly positively correlated with policy subsidy, policy information, and government trust. Additionally, E-RCB is also positively correlated with economic level and housing damage loss and negatively correlated with housing structure risk, house age, and earthquake experience.

Government housing subsidy promotes E-RCB of rural households.

Government Trust promotes E-RCB, the more households that agree “local government’s countermeasures to the earthquake are adequate”, the more likely they are to carry out E-RCB.

Houses with relatively early age and brick–concrete structures have implemented relatively more E-RCB. This means, compared with the past, the seismic performance of newly built rural houses has been significantly improved, which reflects the effect of the times; ERHSPP for rural residences have achieved tangible results.

Households with lower economic levels have adopted relatively few E-RCB. Based on this, it is speculated that the obstacle for households to put their wishes for E-RCB into practice is likely to be a lack of funds.

Simple earthquake experience cannot significantly promote the specific E-RCB but shows a significant negative correlation. In terms of earthquake damage experience, what significantly promotes E-RCB is the actual housing loss caused by earthquakes.

The local government’s propaganda and education on local earthquake risk and anti-seismic fortification knowledge will promote residents’ earthquake preparedness.

## 5. Conclusions and Suggestions

China is a developing country, where a large number of residents live in earthquake-hazardous areas. The local government can promote household disaster preparedness knowledge in ways that are acceptable to the local vulnerable groups. The disaster preparedness propaganda can be done through a variety of methods and mediums, (e.g., print, audio, video) translated into the local language that is easy to understand [29].

The Chinese government can provide policy subsidies, which are graded based on family income and provide more subsidies to poor families for housing special subsidies in rural earthquake-hazardous areas, helping families with financial difficulties build affordable earthquake-resistant houses. This discovery is similar to Wu’s conclusion, when policy subsidies are not graded based on family income and provide more subsidies to poor families. The result is often that families with higher economic affordability are more likely to receive policy subsidies, and poor families cannot use the policy, because they cannot afford out-of-pocket expenses after policy subsidy reduction [33].

Timing of housing seismic supervision: The Chinese government may pay attention to the seismic fortification supervision in the process of building new houses in rural areas, and in the restoration and reconstruction after natural disasters, so that newly built rural houses meet the local seismic fortification standards.

The local government should continue to publish unified, earthquake-resistant housing design drawings, which are suitable for local architectural styles in the local disaster reduction plan.

The local government can conduct more earthquake-resistant technical training for rural craftsmen and encourage welcoming NGOs and community residents to participate in craftsman training and disaster education.

## Figures and Tables

**Figure 1 ijerph-17-09079-f001:**
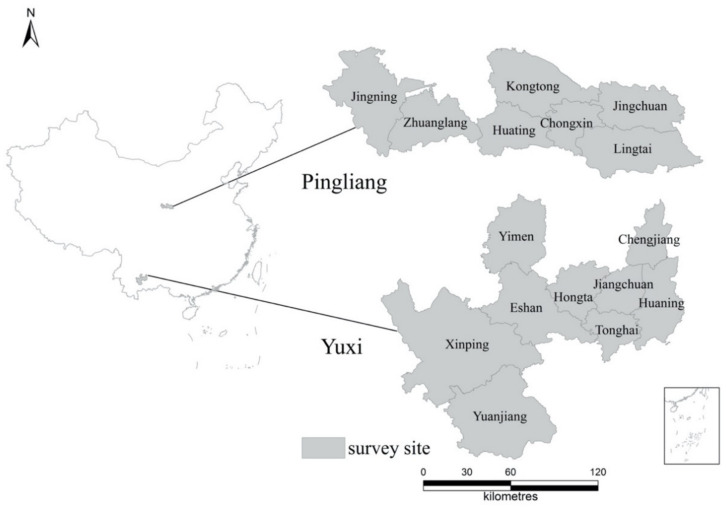
Location of the survey site.

**Figure 2 ijerph-17-09079-f002:**
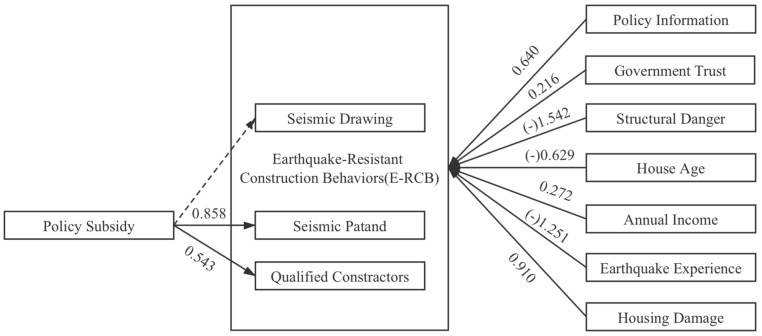
Model.

**Table 1 ijerph-17-09079-t001:** The situation of the survey site.

Sampling Area	Seismic Fortification Intensity	Survey Year	Average Rural Annual Income	Geographical Division	N
Pingliang	7–8	2010	3136 yuan	North-west	1169
Yuxi	7–8	2011	6616 yuan	South-west	1501

**Table 2 ijerph-17-09079-t002:** Descriptive statistics of interviewee. (N = 2670).

Variable		N%	Variable		N%
Gender	Female	39.70%	Age	15–18	0.26%
	Male	59.96%		18–30	8.31%
Education	Unschooled	23.75%		31–40	25.96%
	Primary	38.46%		41–50	29.25%
	Junior	29.10%		51–60	19.25%
	Senior	7.53%		61–70	11.57%
	College and above	0.79%		71–85	5.32%

**Table 3 ijerph-17-09079-t003:** Descriptive statistics (N = 2670).

Variable		N%	Mean	SD
**Dependent Variable**				
Policy Subsidy Received the government housing subsidy when building this house with seismic structure.			0.111	0.314
	Yes (1)	6.03%		
	No (0)	48.28%		
Earthquake-Resistant Construction Behaviors (E-RCB) The number of completed measures of seismic drawing, seismic patands, and qualified contractors.			0.650	0.947
	0 (0)	65.81%		
	1 (1)	5.54%		
	2 (2)	9.78%		
	3 (3)	16.93%		
**Independent variable**				
Seismic Drawing Used seismic drawing when building this house.			0.168	0.374
	Yes (1)	15.88%		
	No (0)	78.61%		
Seismic Patands Made seismic patands when building this house.			0.335	0.472
	Yes (1)	27.75%		
	No (0)	55.09%		
Qualified Contractors Chosen qualified contractors with seismic construction skills when building this house.			0.099	0.298
	Yes (1)	9.48%		
	No (0)	86.55%		
Policy Information Knew the seismic fortification intensity of the area (or the seismic fortification standard to be achieved when building houses).			0.103	0.304
	Yes (1)	10.15%		
	No (0)	88.69%		
Government Trust Recognition degree of “the local government’s earthquake countermeasures are adequate”.			1.788	0.947
	Inconsistent (0)	10.67%		
	Not very consistent (1)	20.56%		
	Relatively consistent (2)	37.42%		
	Consistent (3)	22.58%		
Structural Danger Risk of the building structure			1.208	0.697
	Ring beam/steel frame (0)	13.18%		
	Brick-wood (1)	38.61%		
	Adobe (2)	30.22%		
House Age In (the survey year—the house construction year)			2.400	1.043
Annual Income (RMB)			3.814	1.834
	<5k (1)	14.53%		
	5k–8k (2)	10.34%		
	8k–10k (3)	12.32%		
	10k–20k (4)	27.08%		
	20k–30k (5)	15.51%		
	30k–50k (6)	10.52%		
	50k–80k (7)	4.16%		
	80k–100k (8)	0.67%		
	100k–200k (9)	0.97%		
	200k–500k (10)	0.26%		
	>5000k (11)	0.07%		
Earthquake Experience Experienced 1970 Tonghai or 2008 Wenchuan earthquake.			0.554	0.497
	Yes (1)	55.06%		
	No (0)	44.34%		
Housing Damage House has collapsed due to an earthquake.			0.320	0.467
	Yes (1)	30.19%		
	No (0)	64.19%		

**Table 4 ijerph-17-09079-t004:** Logistic regression results of policy subsidy and ECRB.

Independent Variable	Model A	Model B	Model C	Model D
Seismic Drawing	0.615 ***			−0.177
	(0.197)			(0.257)
Seismic Patand		0.830 ***		0.858 ***
		(0.184)		(0.214)
Qualified Constractors			0.765 ***	0.543 *
			(0.246)	(0.291)
Cons.	−2.179 ***	−2.349 ***	−2.141 ***	−2.349 ***
N	1398	1123	1412	1070
Log-likelihood	−490.742	−406.301	−494.688	−393.316
Pseudo R2	0.009	0.024	0.009	0.030

*** *p* < 0.01, * *p* < 0.1.

**Table 5 ijerph-17-09079-t005:** Ordered logistic regression results of E-RCB.

Independent Variable	Model 1	Model 2	Model 3	Model 4
Policy Information	0.664 ***	1.006 ***	0.717 ***	0.657 ***
	(0.138)	(0.164)	(0.170)	(0.174)
Government Trust	0.126 ***	0.142 **	0.115 *	0.216 ***
	(0.048)	(0.058)	(0.061)	(0.065)
Structural Danger		−1.405 ***	−1.369 ***	−1.542 ***
		(0.099)	(0.104)	(0.114)
House Age		−0.545 ***	−0.623 ***	−0.627 ***
		(0.062)	(0.065)	(0.067)
Annual Income			0.296 ***	0.278 ***
			(0.033)	(0.034)
Earthquake Experience				−1.247 ***
				(0.170)
Housing Damage				0.913 ***
				(0.172)
Cut 1	0.727 ***	−1.995 ***	−1.009 ***	−1.471 ***
	(0.100)	(0.173)	(0.210)	(0.237)
Cut 2	1.739 ***	−0.679 ***	0.397 *	0.013
	(0.107)	(0.166)	(0.210)	(0.234)
Cut 3	2.858 ***	0.624 ***	1.702 ***	1.382 ***
	(0.127)	(0.174)	(0.219)	(0.245)
N	1897	1535	1497	1426
Log-likelihood	−2022.052	−1417.215	−1334.361	−1242.050
Pseudo R2	0.007	0.172	0.200	0.216

*** *p* < 0.01, ** *p* < 0.05, * *p* < 0.1.

**Table 6 ijerph-17-09079-t006:** Summary of support for hypotheses.

Hypothesis	Statement	Support
**H1a**	Government project effect and E-RCB are positively correlated.	Supported
**H1b**	Government project effect and E-RCB are negatively correlated.	Not supported
**H1c**	Government housing subsidy and E-RCB are positively correlated.	Supported
**H1d**	Government housing subsidy and E-RCB are negatively correlated.	Not supported
**H2**	Structural danger and house age are negatively related to E-RCB.	Supported
**H3**	Economic level and E-RCB are positively correlated.	Supported
**H4**	Earthquake experience and E-RCB are positively correlated.	Partially supported

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
