# Peer review of "An Analysis of Rural Households’ Earthquake-Resistant Construction Behavior: Evidence from Pingliang and Yuxi, China"

_ijerph, 2020, doi:10.3390/ijerph17239079_

Round 1

Reviewer 1 Report

The topic of the paper is interesting as it was directed towards evaluating a potentially successful post-earthquake rehabilitation and reconstruction project in China. But the paper has serious flaws which should be considered for rigorous examination before it would be consider for the publication. Title : The title is too long and difficult to understand. Abstract : Very difficult to comprehend. Please clearly state - research problem, the purpose and objectives of the study and your key findings. At this moment it is utterly confusing what you want to achieve. Line 21 : “We studied policy and non-policy factors on Earthquake-Resistant 21 Construction Behaviors (E-RCB) by the logistic regression model.” I am not sure about the “policy” and “non-policy” factors. This is not an established terminology in DRR or in policy science. Introduction : Line 60 – 64 : “Prior research on household disaster preparedness was often limited to psychological aspects and adaptive behaviors while ignoring the policies' effect. This paper adopts ordered logistic regression to quantitatively analyze E-RCB, ERHSPP effect, and other aspects to investigate: 1) Policy subsidy promoted rural household E-RCB? 2) Policy effect, house risk, economic level, and earthquake experience are related to E-RCB?” This is contradictory with the abstract. How do you differentiate between social and psychological factirs for policy factors ? What do you mean by policy factors ? There is no single line dedicated on this topic. Line 65 – 157 : Then you suddenly jumped into another domain. On the one hand you are are talking about policy questions, but you would like to connect the study with risk perception. I am not sure how these variables are connected with policy and your reconstruction works. Whereas, you argued that your research is different from social and cultural dimension of BBB. This is totally contradictory and misleading. Unless the purpose and objectives of the study are clear, the paper is not eligible for further review and consideration for publication.

Author Response

Point 1: The title is too long and difficult to understand. 

Response 1- We have simplified the Title to "An Analysis of Rural Households’ Earthquake-Resistant Construction Behavior: Evidence from Pingliang, and Yuxi, China". 

Point 2:  Abstract -Very difficult to comprehend. Please clearly state - research problem, the purpose and objectives of the study and your key findings. At this moment it is utterly confusing what you want to achieve. 

Response 2-We revised the abstract as research questions, purpose and main findings in Line 18-28. The details are as follows “We studied on Earthquake-Resistant Construction Behaviors (E-RCB) by the logistic and the ordered logistic regression models. Results show that (1) Government housing subsidy promotes E-RCB of rural households. (2) E-RCB was affected by ERHSPP, positively correlated with economic status and housing earthquake damage. (3) E-RCB was negatively correlated with structure danger, house age, and earthquake experience. (4) Housing earthquake damage, not earthquake experience, strikingly promoted E-RCB in rural China. The results could provide suggestions in communication risk strategies for the government. We suggest the local government should promote local acceptable disaster propaganda, provide hierarchical housing subsidies, pay attention to housing seismic supervision, publish earthquake-resistant housing design drawings, and conduct more earthquake-resistant technical training for rural craftsmen.” 

Point 3: Line 21: “We studied policy and non-policy factors on Earthquake-Resistant 21 Construction Behaviors (E-RCB) by the logistic regression model.” I am not sure about the “policy” and “non-policy” factors. This is not an established terminology in DRR or in policy science.  

Response 3- Policy factors are important issues in DRR. Here we discussed a specific policy ERHSPP in the field of earthquake preparedness in disaster reduction in China, and whether it is significantly related to E-RCB. This is just a small empirical analysis. Following your kind suggestions, we divided the influencing factors into four categories: government project effect, house risk, economic level, and earthquake experience. 

Point 4: Line 60 – 64 : “Prior research on household disaster preparedness was often limited to psychological aspects and adaptive behaviors while ignoring the policies' effect.  

Response 4-We revised the sentence as “Prior research on household disaster preparedness discussed psychological aspects and adaptive behaviors. We would like to explore the factors affecting actual behaviors.” 

Point 5: This paper adopts ordered logistic regression to quantitatively analyze E-RCB, ERHSPP effect, and other aspects to investigate: 1) Policy subsidy promoted rural household E-RCB? 2) Policy effect, house risk, economic level, and earthquake experience are related to E-RCB?” This is contradictory with the abstract. How do you differentiate between social and psychological factirs for policy factors? What do you mean by policy factors? There is no single line dedicated on this topic. Line 65 – 157 : Then you suddenly jumped into another domain. On the one hand you are talking about policy questions, but you would like to connect the study with risk perception. I am not sure how these variables are connected with policy and your reconstruction works. 

Response 5- We revised the abstract and Line 96-136, We re-explained the literature related to government project effect, indeleted some content and revised the hypothesis.   

Point 6: Whereas, you argued that your research is different from social and cultural dimension of BBB. This is totally contradictory and misleading. Unless the purpose and objectives of the study are clear, the paper is not eligible for further review and consideration for publication.   

Response 6- We re-defined the research goals in the manuscript based on your suggestions, and thank you for your corrections. Line 63-72, “This paper adopts logistic and ordered logistic regression to quantitatively analyze E-RCB, ERHSPP effect, and other aspects to investigate:(1) Government housing subsidy promoted rural household E-RCB, or not? (2) Government project effect, house risk, economic level, and earthquake experience are related to E-RCB, or not?” 

Reviewer 2 Report

This comprehensive study unfolds different aspects of knowledge, perception and government efforts towards resilient community. Few comments and suggestions are listed after thoroughly reviewed the paper.

  • More explanations on whom the questionnaire asked? (anyone form household or head of the household)
  • Literacy level of the participant would better to present as it may depends on the knowledge.
  • Please check the data provided in Table 2 (Total percentage of responses for the question ‘Qualified contractors’ is more than 100%!).
  • There are lack of responses for different questions tabulated descriptive statistics in Table 2! cumulative percentage is less than 100 except in one(where it is more than 100 as mentioned in no 3). Have you used different questionnaire or any other reason for this?
  • Authors have studied two different areas, experienced large earthquakes. However the date of large earthquakes were very much different (1970 and 2008). When looking at the result, experience of earthquake disaster is very much varying (about 55% says Yes). Considering the results variation, it is suggested to provide location specific results and the combined one.
  • Authors discussed about the perception, earthquake experience and awareness; they might look at this article https://geoenvironmental-disasters.springeropen.com/articles/10.1186/s40677-020-00150-2 which presents a brief on post Gorkha earthquake 2015 scenario how people changed their behavior.

Author Response

Point 1: Few comments and suggestions are listed after thoroughly reviewed the paper.

Response 1-We added suggestions from Line 349 to Line 371.

Point 2: More explanations on whom the questionnaire asked? (anyone form household or head of the household)

Response 2- From Line 186 -192 and Table 2, we added the corresponding description to explain the literacy level of the participant. Also, we added statistical information on the knowledge level of the interviewees , and described the survey and face-to-face interview method used in the survey.

Point 3: Literacy level of the participant would better to present as it may depends on the knowledge. Please check the data provided in Table 2 (Total percentage of responses for the question ‘Qualified contractors’ is more than 100%!).There are lack of responses for different questions tabulated descriptive statistics in Table 2! cumulative percentage is less than 100 except in one(where it is more than 100 as mentioned in no 3). Have you used different questionnaire or any other reason for this?

Response 3-Data percentage check: 86.55% in Table 3 (Table 2 in the previous version) has been confirmed to be an error in data verification. Thank you for your insight!

Point 4: Authors have studied two different areas, experienced large earthquakes. However the date of large earthquakes were very much different (1970 and 2008). When looking at the result, experience of earthquake disaster is very much varying (about 55% says Yes). Considering the results variation, it is suggested to provide location specific results and the combined one.

Response 4- The major earthquakes in 1970 and 2008: Line 197-210, we have added a description. At midnight on December 3, 1970, a 5.5Ms earthquake occurred in Xiji. As residents mainly live in adobe arch kilns with extremely poor earthquake resistance, serious disasters have been caused. In this earthquake, 117 people were killed and 1,540 cave houses collapsed. The disaster area was about 540 km². Pingliang City was affected. This corresponds to the 1970 Tonghai earthquake in Yuxi. In the Wenchuan Earthquake in 2008, both places suffered losses due to weak housing. Considering that it was a long time since the survey in 1970, we asked about the relatively recent Wenchuan Earthquake in Pingliang's questionnaire. We are writing a paper discussing the differences between the two places, thank you for your careful attention to this difference.

Point 5: Authors discussed about the perception, earthquake experience and awareness; they might look at this article https://geoenvironmental-disasters.springeropen.com/articles/10.1186/s40677-020-00150-2 which presents a brief on post Gorkha earthquake 2015 scenario how people changed their behavior.

Response 5-Added references: We have read the paper recommended by you and its references in detail, and learned great new ideas. We have added content to the literature review and suggestions, please see Line 108-111, Line 351 -354.

Reviewer 3 Report

It is interesting that this paper focuses on the influence of policy and non policy factors on seismic behavior of buildings. The review of literature, which can be found in the Introduction section, is complete and pertinent. Moreover, the figures, tables and equations are relevant and fit with the explanation. The results and discussions are justified and logical. The paper is interesting, and it is a good contribution to the knowledge transfer between research institutions and society. However, there are some comments that need to be addressed.

(1).This paper claims that " (1) E-RCB was affected by ERHSPP, positively correlated with economic status and housing earthquake damage, and negatively correlated with structure danger, house age, and earthquake experience. (2) ERHSPP has worked, and the most effective way may lie in the propaganda of earthquake fortification knowledge". Whether the seismic technology level is advanced or not, is it the influencing factor?

(2)Authors claims that "The logistic regression results of Model A-C show that policy subsidy is positively correlated with the three independent input E-CRB variables".Are there any other independent variables?

Author Response

Point 1: This paper claims that " (1) E-RCB was affected by ERHSPP, positively correlated with economic status and housing earthquake damage, and negatively correlated with structure danger, house age, and earthquake experience. (2) ERHSPP has worked, and the most effective way may lie in the propaganda of earthquake fortification knowledge". Whether the seismic technology level is advanced or not, is it the influencing factor?

Response 1- We are very pleased that you have proposed new ideas for this research. Taking into account the economic level of rural households in China, scientists and engineers usually consider affordable and lower-cost housing projects for families, and use relatively inexpensive seismic technology. The seismic technology of these rural houses may not be the most advanced, but practical measures that the residents can affordable. We are happy to use seismic technology as a research variable in future comparative studies.

Point 1: Authors claims that "The logistic regression results of Model A-C show that policy subsidy is positively correlated with the three independent input E-CRB variables".Are there any other independent variables?

Response 2-Other independent variables of model A-C: We have modified the description of the model and data in Line 277-287, welcome to review. The topic of policy subsidies only appears in Yuxi’s questionnaire. We separately analyzed this part of Yuxi’s data. This is a shortfall based on facts. We will continue to focus on other independent variables in future research.

Round 2

Reviewer 1 Report

I am happy with the response authors made against my previous comments.